# Endoscopic Ultrasound Plus Endoscopic Retrograde Cholangiopancreatography Based Tissue Sampling for Diagnosis of Proximal and Distal Biliary Stenosis Due to Cholangiocarcinoma: Results from a Retrospective Single-Center Study

**DOI:** 10.3390/cancers14071730

**Published:** 2022-03-29

**Authors:** Edoardo Troncone, Fabio Gadaleta, Omero Alessandro Paoluzi, Cristina Maria Gesuale, Vincenzo Formica, Cristina Morelli, Mario Roselli, Luca Savino, Giampiero Palmieri, Giovanni Monteleone, Giovanna Del Vecchio Blanco

**Affiliations:** 1Gastroenterology Unit, Department of Systems Medicine, University of Rome “Tor Vergata”, 00133 Rome, Italy; edoardo.troncone@ptvonline.it (E.T.); fabio.gadaleta.11@gmail.com (F.G.); omeroalessandro.paoluzi@ptvonline.it (O.A.P.); cristinagesuale@gmail.com (C.M.G.); gi.monteleone@med.uniroma2.it (G.M.); 2Medical Oncology Unit, Department of Systems Medicine, University of Rome “Tor Vergata”, 00133 Rome, Italy; vincenzo.formica@ptvonline.it (V.F.); cristina.morelli@ptvonline.it (C.M.); mario.roselli@ptvonline.it (M.R.); 3Pathology Unit, Department of Biomedicine and Prevention, University of Rome “Tor Vergata”, 00133 Rome, Italy; luca.savino@ptvonline.it (L.S.); giampiero.palmieri@uniroma2.it (G.P.)

**Keywords:** biliary neoplasia, brushing cytology, cholangiocarcinoma, EUS-FNB, ERCP

## Abstract

**Simple Summary:**

The diagnosis of cholangiocarcinoma depends on several factors, including growth pattern and location. Previous studies have evaluated the diagnostic accuracy of endoscopic retrograde cholangiopancreatography based tissue sampling and endoscopic ultrasound with either fine-needle aspiration or fine-needle biopsy, reporting values < 80% for each procedure. Here, we describe the performance of both methods in a group of patients with a stricture of the biliary tract suspicious for cholangiocarcinoma. Our analysis confirms the high diagnostic accuracy of the procedures when performed together in distinguishing between a primary malignant or benign biliary stenosis.

**Abstract:**

Differentiating between benign and malignant biliary stenosis (BS) is challenging, where tissue diagnosis plays a crucial role. Endoscopic retrograde cholangiopancreatography (ERCP)-based tissue sampling and endoscopic ultrasound (EUS) with fine-needle aspiration (FNA) or biopsy (FNB) are used to obtain tissue specimens from BS. The aim of this retrospective study was to evaluate the diagnostic yield of EUS-FNA/B plus ERCP with brushing or forceps biopsy in BS. All endoscopic procedures performed in patients with BS at our gastroenterology unit were reviewed. The gold standard for diagnosis was histopathology of surgical specimens or the progression of the malignancy at radiological or clinical follow-up. A total of 70 endoscopic procedures were performed in 51 patients with BS. Final endoscopic diagnosis was reached in 96% of the patients and was malignant in 61.7% and benign in 38.3% of cases. Sensitivity, specificity, and diagnostic accuracy were 73.9%, 100%, and 80%, respectively, for EUS-FNA/B; 66.7%, 100%, and 82.5% for ERCP; and 83.3%, 100%, and 87.5% for both procedures carried out in the same session. The combination of EUS and ERCP tissue sampling seems to increase diagnostic accuracy in defining the etiology of BS. Performing both procedures in a single session reduces the time required for diagnostic work-up and optimizes resources.

## 1. Introduction

Biliary stenosis (BS) is a frequent, common, clinical condition in patients hospitalized for jaundice, caused by a narrowing of the biliary tree due to several conditions. Malignant BS (MBS) often results from pancreatic adenocarcinoma or cholangiocarcinoma (CCA). CCA is a tumor arising from the epithelial cells lining the biliary tree and is the second most frequent primary liver malignancy worldwide. From an anatomical point of view, CCA is classified as either intrahepatic cholangiocarcinoma (iCCA, involving second-order bile ducts); perihilar cholangiocarcinoma (pCCA, involving the right or left hepatic duct or their junction), also called Klatskin tumor; or distal CCA (involving the common bile duct). Based on macroscopic growth patterns, CCA can be further classified as either mass-forming (presence of a nodular lesion in the hepatic parenchyma), periductal-infiltrating (tumor growth inside the duct wall and spread longitudinally along the wall), or intraductal (polypoid or papillary tumor growth toward the duct lumen) [1,2,3]. The Bismuth-Corlette classification is routinely used for the clinical assessment of biliary tree involvement in pCCA; the extent of the disease is described according to biliary tract infiltration (involvement of the common bile duct, confluence, right or left hepatic ducts, and both ducts) [4]. Other causes of MBS are hepatocellular carcinoma, ampullary and gallbladder carcinoma, and liver or regional lymph node metastases. Benign BS (BBS) accounts for up to 30% of all BS and is generally related to iatrogenic biliary damage after liver transplant or cholecystectomy. Less common causes of BBS include chronic pancreatitis, autoimmune pancreatic or biliary disease, and inflammatory stenosis of the papilla [5].

Differentiating between MBS and BBS is crucial, as a misdiagnosis of BS leads to unnecessary surgery or delay in the treatment of malignant strictures. Despite recent improvements in imaging and endoscopic techniques, differential diagnosis of BS still represents a clinical challenge, with indeterminate BS (IBS) accounting for up to 20% of all BS [6]. IBS is defined as BS without a definite tissue, radiological, or clinical diagnosis of benign or malignant disease after exploiting all available methods to determine the specific etiology of the biliary disease [6]. We recently proposed a diagnostic work-up of BS according to the site and features of the lesion [5]. Tissue diagnosis plays a pivotal role in identifying the cause of BS and in classifying the histological type of the biliary tumor. Due to the high rate of recurrence after surgery and adjuvant systemic treatment [7], tissue collection may also be useful in identifying factors predicting the recurrence of the disease and to tailor treatment.

The location of BS is crucial to choosing the most accurate method to obtain tissue for differential diagnosis between MBS and BBS, as well as the optimal time sequence in which to use the diagnostic tools available. Tissue sampling may be accomplished by endoscopic retrograde cholangiopancreatography (ERCP)-based tissue collection with brushing or biopsy and endoscopic ultrasound (EUS) with fine-needle aspiration (FNA) or fine-needle biopsy (FNB). When used in the same or sequential sessions, these endoscopic procedures are used to study the site and extent of the stenosis in two different ways, reveal the presence of liver or pancreatic nodules suspicious for malignancy, and obtain tissue for histopathological diagnosis. Following tissue sampling, biliary drainage can be performed by positioning a plastic or metallic stent during ERCP or during EUS when ERCP is not feasible or in patients with hilar obstruction [8]. 

Currently, the overall diagnostic yield of ERCP-based histopathological diagnosis is estimated to reach a maximum of 75% [5,9], whereas pooled EUS-FNA sensitivity is 80% for the diagnosis of malignancy in the biliary tree [10]. In a systematic review and meta-analysis, De Moura et al. [11] reported that EUS-FNA achieved higher sensitivity, specificity, and accuracy than ERCP-based tissue sampling (76%, 100%, and 94.5% vs. 58%, 98%, and 78.1%, respectively) in a sample of 497 patients with BS. A multicenter study involving 263 patients revealed a significantly higher diagnostic accuracy of EUS-FNA compared with ERCP, although the diagnostic performance of EUS-FNA was better than that of ERCP for pancreatic masses but not for biliary lesions [12]. These findings may be affected to some extent by the heterogeneity of patients, as they derive from studies frequently combining primary BS and BS secondary to pancreatic tumors.

The aim of our study was to evaluate the diagnostic yield of EUS-FNA/B and ERCP-based tissue sampling in primary strictures due to cholangiocarcinoma involving the proximal or distal biliary tract.

## 2. Materials and Method 

### 2.1. Patient Selection and Endoscopic Procedures

All patients referred to our gastroenterology unit with jaundice due to BS suspicious for CCA and who underwent ERCP or EUS from November 2015 to September 2021 were considered eligible for the study. The suspicion of BS was based on clinical history and investigations, including abdominal computed tomography (CT) and magnetic resonance imaging (MRI). The exclusion criteria were previous evidence of pancreatic head mass, a clearly protruding-type ampullary cancer, or the lack of adequate follow-up. The diagnostic work-up for each patient was defined by a multidisciplinary team, including gastroenterologists, oncologists, radiologists, surgeons, and pathologists.

All ERCP examinations were performed using a side-viewing scope (TJF 180, Olympus, Tokyo, Japan). Cholangiography and endoscopic sphincterotomy were carried out after cannulation and before tissue sampling. Endoscopic transpapillary tissue sampling was performed with forceps (FB26N-1, Olympus), and at least three samples were collected. Brushing cytology was achieved with at least 10 passes of a brush (RX Cytology Brush; Boston Scientific, Marlborough, MA, USA) on guidewire in the bile duct. Forceps biopsy and brushing were performed after biliary duct dilatation, if indicated, in the last year of the study period. After tissue sampling, endoscopic biliary stenting was carried out. EUS was performed with a linear array echoendoscope equipped with a 7.5 MHz transducer (GF-UCT 180, Olympus). Tissue sampling was carried out using EUS-FNA with a 22-gauge needle (Expect^TM^, Boston Scientific) or EUS-FNB (Acquire^TM^, Boston Scientific; Shark Core^TM^ FNB, Medtronic, Minneapolis, MN, USA). The needle was moved back and forth at least 10 times without a syringe. At least three passes were performed for each patient. In the last five years of the study period, we used only FNB to collect tissue samples during EUS. The tissue collected during ERCP and EUS was immediately put in formalin, and the adequacy of the material was evaluated in all patients by gross inspection by the operator, as rapid on-site evaluation (ROSE) was not available. All brushing samples were sent for cytological examination in special tubes with fixative liquid. In the case of procedures carried out on the same day, EUS with tissue sampling always preceded ERCP. Cytological and histological findings were defined by the pathologist as either negative, positive for malignancy, or nonconclusive due to inadequacy of the tissue sample. In the case of strong suspicion of malignancy and nonconclusive histopathological findings, the EUS or ERCP procedure was repeated. The gold standard for diagnosis was the histopathology of surgical specimens or clinical outcomes (i.e., death from the disease or evidence of disease progression by CT scan or MRI). Patients with no neoplastic findings from the endoscopic procedures were followed for at least one year. Any complications during or following the procedures were recorded. 

### 2.2. Statistical Analysis

To evaluate the performance of EUS-FNA/B and ERCP tissue sampling in the differential diagnosis of BS, the sensitivity, specificity, and diagnostic accuracy and their 95% confidence intervals (95% CIs) were calculated based on the final gold-standard diagnosis. Parameters of diagnostic yield were calculated for each endoscopic procedure and for the two procedures performed in combination, both at first attempt and after repetition. BS site and repetition of the same procedure were considered factors influencing the performance of EUS-FNA/B and ERCP tissue sampling. Statistical analysis was calculated using MedCalc software (MedCalc Software, Ostend, Belgium).

## 3. Results 

### 3.1. Study Population, Endoscopic Procedures, and Diagnosis

During the study period, 51 patients with BS underwent endoscopic evaluation with EUS or ERCP. We excluded 4 patients because of insufficient follow-up; the analysis of the results therefore refers to 47 patients (28 men (59.6%), median age 73 years (range 49–94)). The characteristics of the patients and procedures are summarized in Table 1. Cross-sectional images showed distal BS in 31 patients (66%) and proximal or hilar BS in 16 patients (34%). 

A total of 70 endoscopic procedures with tissue sampling for histological and cytological analysis were carried out: ERCP with brushing/biopsy in 40 patients and EUS-FNA/B in 30 patients (EUS-FNA in 4 patients and FNB in 26 patients). Twenty-four patients underwent ERCP with brushing/biopsy only, fourteen patients underwent EUS-FNA/B only, and sixteen patients underwent both ERCP with brushing/biopsy and EUS-FNA/B performed in the same session. Six patients with nonconclusive histological and cytological diagnoses repeated the endoscopic procedure: three patients underwent two ERCPs, one patient three ERCPs, and two patients two EUS-FNBs. One case of mild post-ERCP bleeding and one case of mild post-ERCP pancreatitis were reported. No adverse events were reported after EUS-FNA/B.

Malignancy was finally diagnosed in 29 patients (61.7%), while the remaining 18 patients (38.3%) had BBS. Of the 29 patients with malignancy, 27 (93.1%) were diagnosed by ERCP- or EUS-based tissue sampling, while the remaining 2 patients were diagnosed by surgery. Of the two patients with surgical diagnoses, one had previously undergone both ERCP with brushing cytology and transpapillary biopsies and EUS-FNB, while the other had only undergone EUS-FNB. Both of these patients underwent a Whipple procedure after the first nonconclusive histopathological result without repetition of tissue sampling because of a strong clinical and radiological suspicion of MBS, deemed operable at clinical and radiological staging. The histology of surgical specimens revealed an infiltrating distal CCA. A total of 13 patients underwent surgery (11 Whipple procedures for distal and 2 hepatic resections for hilar BS). No surgery was performed for benign disease based on the histology of surgical specimens. After a negative histological and cytological report, a final diagnosis of BBS was confirmed with clinical and radiological follow-up after a minimum of 6 months (median follow-up 26.5 months, range 6–59).

### 3.2. Diagnostic Yield of ERCP- and EUS-FNA/FNB-Based Tissue Sampling

The diagnostic yield of ERCP with brushing/biopsy and EUS-FNA/B of BS is shown in Table 2. 

Single ERCP with brushing/biopsy—On the first attempt, the sensitivity, specificity, and diagnostic accuracy were 63.2% (95% CI: 38.4–83.7%), 100% (95% CI: 79.4–100%), and 80% (95% CI: 63.1–91.6%), respectively. Due to nonconclusive histological findings, ERCP was repeated for four patients repeated (one patient underwent three ERCPs), which revealed MBS in two patients and BBS in the remaining two. Considering the repeated procedures, the overall sensitivity, specificity, and diagnostic accuracy of ERCP with brushing/biopsy were 66.7% (95% CI: 43.0–85.4%), 100% (95% CI: 82.4–100%), and 82.5% (95% CI: 67.2–92.7%), respectively.

Single EUS-FNA/B—On the first attempt, the sensitivity, specificity, and diagnostic accuracy were 72.7% (95% CI: 49.8–89.3%), 100% (95% CI: 54.1–100%), and 78.6% (95% CI: 59.5–91.7%), respectively. Due to nonconclusive histological findings, the procedure was repeated for two patients repeated, which revealed BBS in one patient and MBS in the other. Notably, in the latter patient, we performed an EUS-FNB of perihilar adenopathy after a previous ERCP-based brushing and EUS-FNB of hilar stricture, both of which failed to diagnose malignancy. Considering the repeated procedures, the overall sensitivity, specificity, and diagnostic accuracy of EUS-FNA/FB were 73.9% (95% CI: 51.6–89.7%), 100% (95% CI: 59.1–100%), and 80% (95% CI: 61.4–92.3%), respectively. 

In six patients, we performed EUS-FNB of the primary (biliary tree) and potentially secondary (lymph nodes) sites of the disease, which was positive for malignancy in only one patient. 

Combined ERCP- and EUS-FNA/B-based tissue sampling—Sixteen patients underwent both EUS and ERCP sampling in the same session (Figure 1, Figure 2 and Figure 3). The combined procedures showed a sensitivity, specificity, and diagnostic accuracy of 83.3% (95% CI: 51.6–97.9%), 100% (95% CI: 39.8–100%), and 87.5% (95% CI: 61.6–98.4%), respectively. 

### 3.3. Performance of Endoscopic Procedures According to Location of Biliary Stenosis

Diagnostic yield of ERCP with brushing or biopsy and EUS-FNA or EUS-FNB according to location of BS is shown in Table 3.

Distal BS was observed in 31 patients, all of whom underwent ERCP with brushing or biopsy except 1, who was operated on a few days after EUS-FNB with a definite histological diagnosis of malignancy, without performing ERCP. The sensitivity, specificity, and diagnostic accuracy of ERCP in these cases of distal BS were 69.2% (95% CI: 38.6–90.1%), 100% (95% CI: 73.5–100%), and 84% (95% CI: 63.9–95.5%), rising to 73.3% (95% CI: 44.9–92.2%), 100% (95% CI: 78.2–100%), and 86.7% (95% CI: 69.3–96.2%), respectively, after repetition of the procedure in the case of nonconclusive findings at first attempt. Fourteen of the thirty-one patients with distal BS underwent EUS-FNA/B, which showed a sensitivity, specificity, and diagnostic accuracy of 44.4% (95% CI: 13.7–78.8%), 100% (95% CI: 47.8–100%), and 64.3% (95% CI: 38.4–88.2%), respectively.

Proximal BS was observed in 16 patients, 10 of whom underwent ERCP with brushing/biopsy and 12 of whom underwent EUS-FNA/B. The sensitivity, specificity, and diagnostic accuracy were 50% (95% CI: 11.8–88.2%), 100% (95% CI: 39.8–100%), and 70% (95% CI: 34.7–93.3%) for ERCP and 90.1% (95% CI: 58.7–99.8%), 100% (95% CI: 2.5–100%), and 91.7% (95% CI: 63.9–99.8%) for EUS-FNA/B, respectively. 

## 4. Discussion

CCA is the second most common primary liver tumor, exhibiting poor prognosis and resistance to chemotherapeutic drugs, and is responsible for the majority of BS. pCCA and iCCA represent >90% of all CCA [1,2,3]. CT scan and MRI with magnetic resonance cholangiopancreatography performed before biliary drainage are considered the standard imaging methods for the preoperative assessment of CCA worldwide, as they provide a comprehensive evaluation of the primary tumor, the relationship with adjacent structures, and potential chest and abdominal involvement [2,3]. Currently, peroral cholangioscopy during ERCP is performed as an adjunctive procedure during the preoperative assessment of CCA. It provides a detailed definition of the location and extent of the tumor through the biliary tree, which is useful to ensure a radical resection of the neoplasia with negative margin (R0) and to avoid surgery in advanced disease [13,14].

Tissue sampling histology is necessary to obtain a definitive diagnosis of primary liver cancer and to exclude metastatic localization of other tumors. In the case of the radiological diagnosis of CCA and evidence of surgical resectability, when extrahepatic secondary localization of the disease has been excluded, the patient may undergo surgical resection without a prior histological diagnosis in selected cases. 

In the case of unresectable neoplasia, tissue diagnosis is mandatory for choosing the most appropriate treatment. ERCP-based tissue sampling and EUS-FNA/B are the most common methods used to collect tissue for a differential diagnosis of BS, whether associated with liver mass or not. EUS-FNA/B is the method of choice with distal biliary stenosis due to pancreatic mass [15], which is not approachable with ERCP. Moreover, EUS can provide additional information regarding the presence of biliary mass, the extension of ductal wall thickening, and the presence of regional lymphadenopathy. In contrast, ERCP is the first-choice method to obtain tissue diagnosis of intrahepatic BS by brushing, forceps biopsy, or both.

The diagnostic power of these endoscopic procedures for the diagnosis of malignancy is influenced by the expertise of the operator, the site of the lesion, and the growth pattern of the disease; a wide range of sensitivity values are reported as being 21–72% for ERCP brushing [9,16,17,18], 43–76% for ERCP forceps biopsy [1,9,16,17,19,20], and 43–94% for EUS-FNA [11,12,15,21,22]. Many studies on the diagnostic yield of EUS-FNA/B and ERCP-based tissue sampling for histopathological diagnosis of BS were carried out on heterogeneous populations, including patients with BS due to pancreatic or biliary tumors. In a recent meta-analysis of 61 studies, the sensitivity of tissue sampling obtained with ERCP biopsy, ERCP brushing plus biopsy, and EUS-FNA in patients with pCCA and distal CCA was 67%, 70.7%, and 73.6%, respectively [23].

In our study, we excluded patients with BS secondary to pancreatic cancer in order to analyze data relating to a more homogeneous population. Overall, the sensitivity of EUS-FNA/B was slightly higher than that of ERCP-based sampling, considering both brushing and forceps biopsy tissue acquisition (73.9% vs. 66.7%, respectively). A key finding of this study is that repeating EUS or ERCP-based tissue sampling increases the diagnostic power of both procedures, reducing the risk of referring a non-neoplastic patient to the surgeon for biliary resection. This result is in line with our previous experience [24] and with the European Society of Gastrointestinal Endoscopy (ESGE) guidelines, showing a rise in the diagnostic accuracy of EUS-FNA of up to 90% [15,25,26]. As a factor potentially affecting the diagnostic power of EUS tissue sampling, the type of needle was previously evaluated in a meta-analysis comparing the diagnostic accuracy of EUS-FNA and EUS-FNB in pancreatic and nonpancreatic solid mass; by providing a core tissue sample, EUS-FNB provided a higher pooled diagnostic accuracy compared with EUS-FNA, with fewer passes required to obtain sufficient material for diagnosis [27]. To date, few studies have assessed the efficacy of EUS-FNB in the diagnostic work-up of CCA [28]. In our study, the majority of EUS sampling was performed using FNB with no complications. However, the small number of procedures performed with FNA did not allow for a comparison of the performance of the two needles. 

The location and growth patterns of CCA may affect the performance of endoscopic tissue sampling procedures. In a subanalysis focusing on the location of the disease, we found that ERCP-based tissue sampling had a higher diagnostic yield than EUS-FNA/B in distal BS, in contrast with previous studies [11,12,28]. This discrepancy may be due to the small number of patients with distal BS undergoing EUS-FNA/B, and to the growth pattern of the disease, characterized only by the thickening of the biliary wall without the presence of a clear intraductal mass. EUS-FNA/B showed a higher sensitivity and diagnostic accuracy in proximal BS than ERCP-based tissue sampling. This finding is probably related to the presence of perihilar liver nodules and intraductal growth occluding the lumen of the biliary duct, which makes sampling by EUS-FNA/B easier. 

Our study confirmed that the combination of EUS and ERCP improves diagnostic performance in BS. In a subgroup of 16 patients undergoing both endoscopic procedures, we observed a rise in the diagnostic accuracy of combined procedures of up to 90%. This finding is in line with those of a previous study [12] reporting that the combination of EUS and ERCP led to an increase in diagnostic accuracy of up to 87.1% compared with 76% for EUS-FNA and 60.5% for ERCP when performed separately. Although combined EUS and ERCP tissue sampling in the same session for BS is becoming more common in referral centers [28], its widespread use depends on the training of operators. Performing both ERCP and EUS-FNA/B in the same session should be promoted as it may reduce the time and cost of hospitalization.

The diagnostic and therapeutic complexity of BS due to CCA requires a multidisciplinary approach in tertiary centers equipped with all the diagnostic tools currently available. Over the years, technological advancements have made it possible to obtain more adequate tissue sampling for diagnostic purposes. We hope that in the near future, the development of other approaches, such as the use of biohumoral markers, may facilitate early diagnosis and provide possible predictors of response to therapy.

Our study has several limitations. First, it is a retrospective investigation, even if the data were prospectively recorded. The number of patients is relatively small, limiting a statistically significant comparison between the two procedures in terms of diagnostic accuracy. Only 16 patients underwent combined EUS and ERCP-based tissue sampling, as we only started to routinely perform both procedures in the last two years of the study period. In addition, we began using EUS-FNB instead of EUS-FNA in 2017. We cannot exclude that this change in the diagnostic work-up may have influenced the performance of the endoscopic procedures analyzed here.

A strength of our study is the selection of patients. We evaluated a homogeneous group of patients with jaundice and evidence of BS, excluding cases of BS due to pancreatic cancer or metastatic disease. To the best of our knowledge, very few studies have evaluated the role of ERCP and EUS sampling in patients with a disease involving exclusively the biliary tract. Our specificity was higher with no false positive samples. However, the pathologist was not blinded because all patients were discussed in a multidisciplinary team meeting before the endoscopic procedure. Nevertheless, our study provides useful insights supporting previous evidence regarding the advantage of combining the two endoscopic procedures in the same session to obtain a more rapid, accurate, and definitive histopathological diagnosis. 

## 5. Conclusions

The diagnostic accuracy of EUS-FNA and ERCP tissue sampling in BS is influenced by the growth patterns and the location of the disease, as well as by the presence of liver mass. The combination of EUS-FNB and ERCP-based tissue sampling increases the diagnostic power of each procedure when performed alone. A multidisciplinary approach plays a crucial role in defining neoplastic biliary disease and in tailoring the most appropriate treatment.

## Figures and Tables

**Figure 1 cancers-14-01730-f001:**
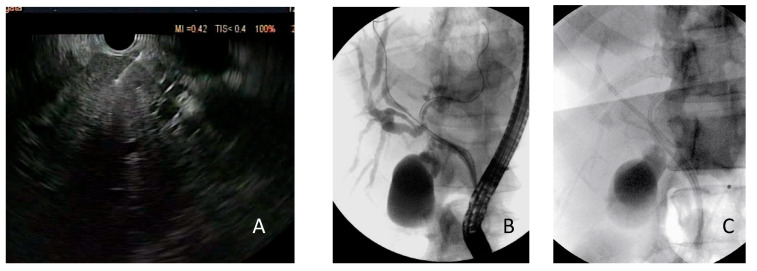
Case of proximal biliary stenosis with liver nodule finally diagnosed as cholangiocarcinoma. (**A**) EUS-FNB of the liver nodule. (**B**,**C**) Biliary drainage performed after the placement of two guidewires (**B**) with plastic stenting (**C**) in the same session as the EUS procedure.

**Figure 2 cancers-14-01730-f002:**
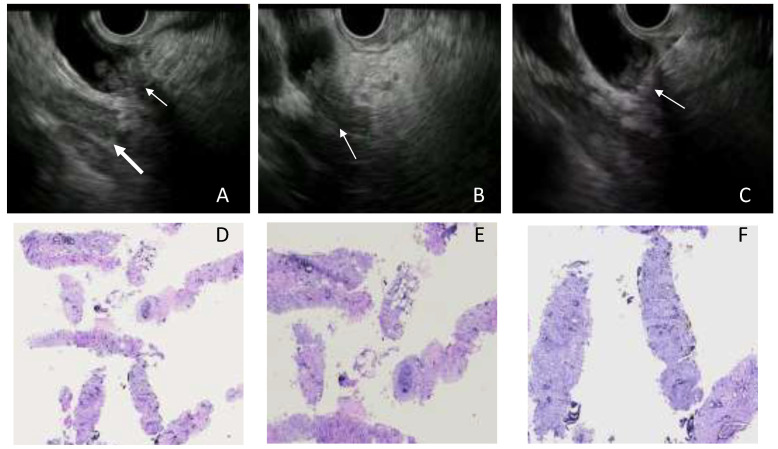
Case of medio-proximal biliary stricture. (**A**–**C**) At EUS, the biliary stenotic tract was clearly visible due to the thickening of the wall and vegetation inside the choledochus ((**A**,**B**) thin arrow); lymphadenopathy was clearly visible (**A**, thick arrow). EUS-FNA of the choledochus was performed to obtain tissue diagnosis (**C**). (**D**–**F**) Tissue samples collected by EUS-FNB stained with hematoxylin and eosin at different magnifications ((**D**) 2×; (**E**) 4×; (**F**) 10×). Perineural invasion is visible in (**E**).

**Figure 3 cancers-14-01730-f003:**
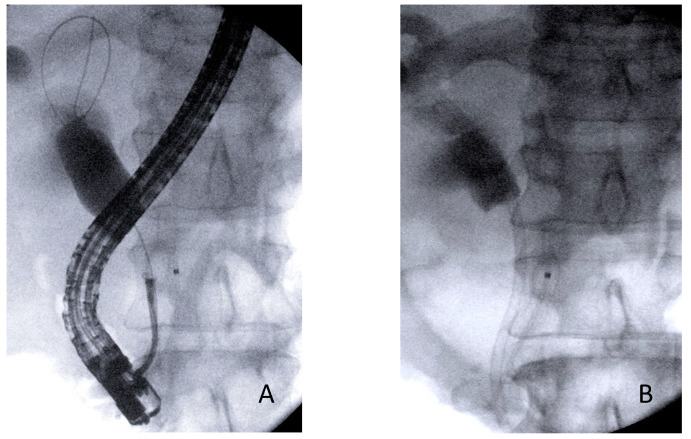
(**A**) Forceps biopsy during ERCP in a patient with medio-proximal biliary stricture. (**B**) At the end of the procedure, biliary drainage was performed with the placement of a plastic stent.

**Table 1 cancers-14-01730-t001:** Characteristics of 47 patients undergoing endoscopic procedures for biliary stenosis.

Sex, Male (%)	28 (59.6)
Age, median (range)	73 (49–94)
Location of stenosis, *n* (%)	
Distal	31 (66)
Proximal	16 (34)
Appearance of stenosis, *n* (%)	
Distal with mass	4 (8.5)
Distal thickening/infiltrating disease	27 (57.4)
Proximal with mass	5 (10.6)
Proximal thickening/infiltrating disease	11 (23.4)
Final diagnosis, *n* (%)	
Malignant	29 (61.7)
Distal	17 (3.2)
Proximal	12 (25.5)
Benign	18 (38.3)
Distal	14 (29.8)
Proximal	4 (8.5)
Endoscopic procedures, *n* (%)	70 (100)
ERCP with brushing/biopsy	40 (57.1)
EUS	30 (42.9)
EUS-FNA	4 (13)
EUS-FNB	26 (87)
Combined ERCP + EUS-FNA/B	16 (22.8)

**Table 2 cancers-14-01730-t002:** Diagnostic yield of ERCP with brushing/biopsy and EUS-FNA/FNB in biliary stenosis.

Procedure	Sensitivity (95% CI)	Specificity (95% CI)	Accuracy (95% CI)
ERCP with brushing/biopsy 1st attempt	63.2% (38.4–83.7)	100% (79.4–100)	80% (63.1–91.6)
ERCP with brushing/biopsy Overall	66.7% (43.0–85.4)	100% (82.4–100)	82.5% (67.2–92.7)
EUS-FNA/B 1st attempt	72.7% (49.8–89.3)	100% (54.1–100)	78.6% (59.5–91.7)
EUS-FNA/B Overall	73.9% (51.6–89.7)	100% (59.1–100)	80% (61.4–92.3)
EUS-FNA/B + ERCP	83.3% (51.6–97.9)	100% (39.8–100)	87.5% (61.6–98.4)

**Table 3 cancers-14-01730-t003:** Diagnostic yield of ERCP with brushing/biopsy and EUS-FNA/B according to site of biliary stenosis.

Procedure	Sensitivity (95% CI)	Specificity (95% CI)	Accuracy (95% CI)
Distal biliary stenosis			
ERCP with	73.30%	100%	86.70%
brushing/biopsy	(44.9–92.2)	(78.2–100)	(69.3–96.2)
EUS-FNA/B			
	44.40%	100%	64.30%
	(13.7–78.8)	(47.8–100)	(38.4–88.2)
Proximal biliary stenosis			
ERCP with	50%	100%	70%
brushing/biopsy	(11.8–88.2)	(39.8–100)	(34.7–93.3)
EUS-FNA/B			
	90.10%	100%	91.70%
	(58.7–99.8)	(2.5–100)	(63.9–99.8)

## Data Availability

The data presented in this study are available on request from the corresponding author. The data are not publicly available due to privacy regulations in force at our hospital.

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
