# Peer review of "Endoscopic Ultrasound Plus Endoscopic Retrograde Cholangiopancreatography Based Tissue Sampling for Diagnosis of Proximal and Distal Biliary Stenosis Due to Cholangiocarcinoma: Results from a Retrospective Single-Center Study"

_cancers, 2022, doi:10.3390/cancers14071730_

Round 1

Reviewer 1 Report

This interesting paper by Troncone and coll. entitled “Endoscopic Ultrasound (Eus) Plus Endoscopic Retrograde Cholangiopancreatography (Ercp)-Based Tissue Sampling In The Diagnosis Of Proximal And Distal Biliary Stenosis Due To Cholangiocarcinoma: Results From A Restrospective Single Center Study” provides a real-life perspective concerning the diagnostic accuracy of endoscopic sampling of biliary strictures

I congratulate to the Authors: the work is clearly written and the study limitations (retrospective study, change of EUS policy and small sample size) are well acknowledged

The english style is fine

I have few recommendations:

1) provide a brief discussion concerning the role of chilangioscopy in preoperative evaluation, possibly referring to this recently published review on CCA management (10.3390/cancers13153657)

2) table 3 is missing, please provide it in the revised version

Best regards

Author Response

We would like to thank the Reviewers for their constructive and helpful comments. We have revised the manuscript taking all the issues raised into account. All changes in the manuscript are highlighted in bold.

Reviewer #1: I have few recommendations:

1) provide a brief discussion concerning the role of cholangioscopy in preoperative evaluation, possibly referring to this recently published review on CCA management (10.3390/cancers13153657)

# We thank the Reviewer for their suggestion. We have now commented on the role of cholangioscopy in preoperative assessment, citing the suggested reference (ref. 13), in the Discussion (page 8 lines 17–21).

2) table 3 is missing, please provide it in the revised version

# We apologize for the omission of table 3, which has now been added (page 7).

Reviewer 2 Report

This is a case series study about tissue sampling for biliary stenosis due to cholangiocarcinoma by ERCP plus EUS. This is a relatively common topic in which there have been some similar studies. I have some questions.

  1. If the authors want to focus on the sensitivity/specificity/accuracy of EUS+ERCP method, they should describe more detail of the 16 cases in which they applied this technique. From previous reports, there are some factors, such as the existence of extraductal mass or the location of the stricture, which can impact on the diagnostic sensitivity/accuracy.
  2. Please describe why the authors did EUS+ERCP in 16 cases, and did not in other cases. Is there any possibility of selection bias?
  3. I feel the combination of EUS and ERCP tissue sampling in the same session for biliary structure is a common strategy. Although EUS seems to have better sensitivity in a recent study (e.g. PMID: 34669743), it is a common practice to do both EUS and ERCP tissue sampling. I am not sure what insight this study can bring.
  4. On page 7, the authors describe they prepare Table 3. However, I do not see any Table3.
  5. Given the small number of cases enrolled, it would be difficult to compare one method with another (e.g. ERCP vs EUS). Of course, there is no statistically significant difference in this study.

Author Response

We would like to thank the Reviewers for their constructive and helpful comments. We have revised the manuscript taking all the issues raised into account. All changes in the manuscript are highlighted in bold.

Reviewer #2:

This is a case series study about tissue sampling for biliary stenosis due to cholangiocarcinoma by ERCP plus EUS. This is a relatively common topic in which there have been some similar studies. I have some questions.

1)         If the authors want to focus on the sensitivity/specificity/accuracy of EUS+ERCP method, they should describe more detail of the 16 cases in which they applied this technique. From previous reports, there are some factors, such as the existence of extraductal mass or the location of the stricture, which can impact on the diagnostic sensitivity/accuracy.

2)         Please describe why the authors did EUS+ERCP in 16 cases, and did not in other cases. Is there any possibility of selection bias?

# We apologize for any lack of clarity. All the 16 patients in question had a disease exhibiting a similar growth pattern. Indeed, this retrospective study refers to a 6-year period, during which time our diagnostic practices changed slightly. In the early years one method was preferred to the other depending on the local availability of operators. In the last two years, we standardized a diagnostic work-up in patients with biliary stenosis including the routine use of both EUS and ERCP-based tissue sampling. We have now clarified these points in the Discussion (page 9, lines 12-21 and 40-42).

3)         I feel the combination of EUS and ERCP tissue sampling in the same session for biliary structure is a common strategy. Although EUS seems to have better sensitivity in a recent study (e.g. PMID: 34669743), it is a common practice to do both EUS and ERCP tissue sampling. I am not sure what insight this study can bring.

# We are grateful to the Reviewer for pointing out this lack of focus. We have now added a sentence in the Discussion, citing the reference suggested (ref. 28), to highlight the importance of combining of EUS and ERCP in the diagnostic work-up of biliary stricture (page 9, lines 27 to 29). We believe that the exclusion of patients with distal biliary strictures due to pancreatic cancer strengthens the role of EUS-FNB in obtaining histological confirmation of biliary disease. The homogeneity of enrolled patients (all with CCA and without pancreatic cancer) increases the relevance of the data emerging from our study; the findings of many previous studies are limited by the heterogeneity of tumors investigated. Furthermore, our work confirms the safety of EUS-FNB, which caused no complications, and the high diagnostic accuracy in the case of benign stenosis. Lastly, our study corroborates the fact that repeating tissue sampling with both EUS-FNB or ERCP increases their diagnostic power. We have now better highlighted these concepts in the Discussion (page 9, lines 46 to 53, page 10 lines 1-2).

4)         On page 7, the authors describe they prepare Table 3. However, I do not see any Table3.

# We apologize for the omission of table 3, which has now been added (page 7).

5)         Given the small number of cases enrolled, it would be difficult to compare one method with another (e.g. ERCP vs EUS). Of course, there is no statistically significant difference in this study.

# We agree with the Reviewer and have now highlighted this limitation of our study more clearly in the Discussion (page 9, lines 39 to 40).

Reviewer 3 Report

Dear Editor, thank you so much for inviting me to revise this manuscript.

This study addresses a current topic.

The manuscript is quite well written and organized. English could be improved.

Figures and tables are comprehensive and clear.

The introduction explains in a clear and coherent manner the background of this study.

We suggest the following modifications:

  • Introduction section: although the authors correctly included important papers in this setting, we believe a couple of studies should be cited within the introduction ( PMID: 35070041 ; PMID: 33571059 ), only for a matter of consistency. We think it might be useful to introduce the topic of this interesting study.
  • Methods and Statistical Analysis: nothing to add.
  • Discussion section: Very interesting and timely discussion. Of note, the authors should expand the Discussion section, including a more personal perspective to reflect on. For example, they could answer the following questions – in order to facilitate the understanding of this complex topic to readers: what potential does this study hold? What are the knowledge gaps and how do researchers tackle them? How do you see this area unfolding in the next 5 years? We think it would be extremely interesting for the readers.

However, we think the authors should be acknowledged for their work. In fact, they correctly addressed an important topic, the methods sound good and their discussion is well balanced.

One additional little flaw: the authors could better explain the limitations of their work, in the last part of the Discussion.

We believe this article is suitable for publication in the journal although major revisions are needed. The main strengths of this paper are that it addresses an interesting and very timely question and provides a clear answer, with some limitations.

We suggest a linguistic revision and the addition of some references for a matter of consistency. Moreover, the authors should better clarify some points.

Author Response

We would like to thank the Reviewers for their constructive and helpful comments. We have revised the manuscript taking all the issues raised into account. All changes in the manuscript are highlighted in bold.

Reviewer #3:

We suggest the following modifications:

1)         Introduction section: although the authors correctly included important papers in this setting, we believe a couple of studies should be cited within the introduction (PMID: 35070041 ; PMID: 33571059) only for a matter of consistency. We think it might be useful to introduce the topic of this interesting study.

# We thank the Reviewer for their suggestions. We have now added a sentence in the Introduction, citing the references suggested (Ref. 7 and 8), to emphasize the role of biopsy not only in diagnosis but also in the potential identification of targeted therapies (page 2, lines 39-42.

# We have also added a sentence, citing the reference suggested (PMID: 35070041), to highlight the therapeutic role of these endoscopic procedures and the possibility of carrying out a biliary drainage by EUS when ERCP fails (page 3, lines 3–5).

2)         Discussion section: Very interesting and timely discussion. Of note, the authors should expand the Discussion section, including a more personal perspective to reflect on. For example, they could answer the following questions – in order to facilitate the understanding of this complex topic to readers: what potential does this study hold? What are the knowledge gaps and how do researchers tackle them? How do you see this area unfolding in the next 5 years? We think it would be extremely interesting for the readers.

 # We added a paragraph describing our prospective for the future in Discussion (page 9, lines 32 to 37)

However, we think the authors should be acknowledged for their work. In fact, they correctly addressed an important topic, the methods sound good and their discussion is well balanced.

# We thank the Reviewer for their positive comments and useful suggestions.

3)         One additional little flaw: the authors could better explain the limitations of their work, in the last part of the Discussion.

# We agree with the Reviewer and have now discussed more clearly the limitations of our study in the Discussion (page 9, lines 38 to 45).

We believe this article is suitable for publication in the journal although major revisions are needed. The main strengths of this paper are that it addresses an interesting and very timely question and provides a clear answer, with some limitations.

We suggest a linguistic revision and the addition of some references for a matter of consistency. Moreover, the authors should better clarify some points.

# English style was carefully revised.

Round 2

Reviewer 1 Report

The revised version is suitable for publication

Reviewer 2 Report

I really appreciate the author's reply and their correction.

1) To describe sensitivity, specificity and accuracy, it would be better to describe concrete numbers (e.g. 14/16), too.

2) If the authors insist EUS-FNA/ERCP combination may be superior, they should present more data to support the idea. I do not see any detail of the 16 patients who had EUS-FNA/ERCP combination biopsies except that they were done for the last 2 years. Given that some factors (as the authors show in Table 3) can impact the accuracy of the biopsies, there is no supportive data in this paper that EUS-FNA/ERCP combination may be superior. 

Reviewer 3 Report

Acceptance.